# Detecting phishing webpages *via* homology analysis of webpage structure

Jian Feng[1], Yuqiang Qiao[1], Ou Ye[1] and Ying Zhang[2]

[1] College of Computer Science & Technology, Xi'an University of Science and Technology, Xi'an, Shaanxi, China

[2] Information Technology Department for Head Office of SPD Bank, National Institute of Standards and Technology Application Development Service Sub-centre (Xi'an), Xi'an, Shaanxi, China



## ABSTRACT

Phishing webpages are often generated by phishing kits or evolved from existing kits. Therefore, the homology analysis of phishing webpages can help curb the proliferation of phishing webpages from the source. Based on the observation that phishing webpages belonging to the same family have similar page structures, a homology detection method based on webpage clustering according to structural similarity is proposed. The method consists of two stages. The first stage realizes model construction. Firstly, it extracts the structural features and style attributes of webpages through the document structure and vectorizes them, and then assigns different weights to different features, and measures the similarity of webpages and guides webpage clustering by webpage difference index. The second phase completes the detection of webpages to be tested. The fingerprint generation algorithm using double compressions generates fingerprints for the centres of the clusters and the webpages to be tested respectively and accelerates the detection process of the webpages to be tested through bitwise comparison. Experiments show that, compared with the existing methods, the proposed method can accurately locate the family of phishing webpages and can detect phishing webpages efficiently.

## INTRODUCTION

Phishing is a kind of social engineering attack, which is a malicious behaviour that deceives network users to visit phishing webpages and attempts to steal various private information (including passwords, bank card numbers, *etc.*) of users. According to the latest report of the APWG (Anti-Phishing Working Group), the total number of phishing webpages in the second quarter of 2020 increased by 17.2% over the same period in 2019 (*APWG, 2020*). The continued growth of phishing attacks has become one of the key factors threatening Internet security, and timely and effective detection of phishing webpages is very important.

In the attack and defence game, phishing webpage detection technologies have been continuously developed, mainly includes traditional and emerging methods. The former includes blacklist-based (*Liang et al., 2016*), heuristic-based (*Xiang et al., 2011*; *Moghimi & Varjani, 2016*), visual similarity (*Raj & Vithalpura, 2018*; *Rao & Pais, 2020*), and

Corresponding author
Jian Feng, fengjian@xust.edu.cn

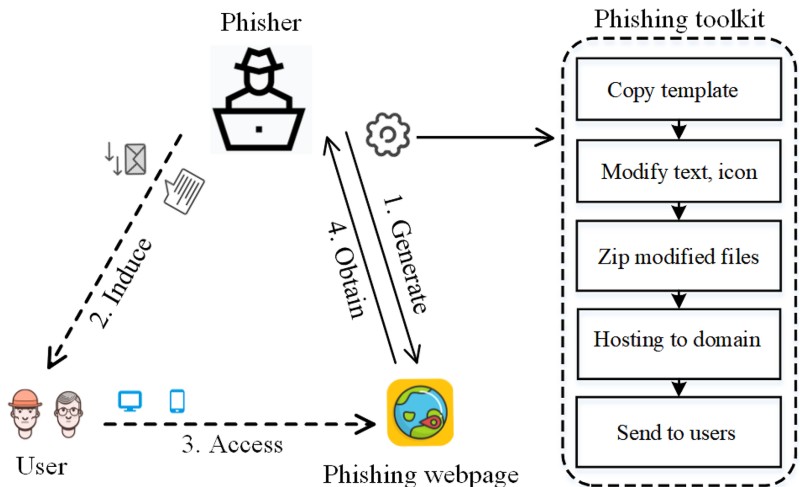

**Figure 1 Typical phishing attack process: (I) phisher uses toolkit to generate phishing webpages; (II) induces users; (III) user accesses phishing webpage; (IV) phisher obtains user's privacy information.**

machine learning-based methods (*Aleroud & Zhou, 2017*; *Rao & Pais, 2019*). And the latter is based on deep learning (*Bahnsen et al., 2017*; *Feng, Zou & Nan, 2019*; *Yang, Zhao & Zeng, 2019*; *Feng et al., 2020*). These detection methods regard phishing webpage detection as a problem of binary classification, by manually or automatically extracting features from URL and content of webpages, and then realizing the identification of phishing webpages through supervised learning models.

However, phishing webpages become complex with the extensive use of various social engineering methods in phishing attacks. This makes more and more difficult to find and extract significant features from webpages, so although the classification-based methods can accurately detect known phishing webpages, but cannot effectively track the source of phishing webpages, so cannot be curbed the proliferation of phishing webpages from the root.

In order to achieve the purpose of fraud quickly and effectively, more than 90% of phishing webpages are automatically generated by phishing webpage generation kits, as shown in Fig. 1; at the same time, to avoid plug-in interception and the higher cost of modifying the template, the newly generated phishing webpages are often gradually evolved from their earlier versions (*Oest et al., 2018*). The phishing webpages generated or evolved by the same kit form a phishing webpage cluster, which has similar characteristics. Therefore, intuitively, tracing the source of phishing webpages through homology analysis would help to find clusters of phishing webpages and effectively prevent attacks.

Based on this point and different from mainstream classification detection methods, this paper regards phishing webpage detection as a clustering problem for homology discovery, hopes to establish the feature model of different categories of phishing webpages, and then guide the detection of phishing webpages by calculating the homology of different categories of phishing webpages to be tested. The key to the paper is the similarity learning of webpages, which is very challenging, because the subtle differences in

text can make two webpages very different semantically, while webpages with different text may still be similar. Therefore, a successful model should: (1) use the structure of webpages instead of text, (2) infer the similarity of webpages from the structural information of webpages, (3) be fast and efficient.

Based on the above analysis, a Structure based Phish Homology Detection Model (SPHDM) is proposed. Firstly, the structural features of webpages are extracted, and the similarity calculation method is proposed to find clusters of the phishing webpages; secondly, an efficient fingerprint algorithm is designed to accelerate the comparison and classification of the webpages to be tested. The results show that SPHDM has fast and effective detection capabilities comparing to clustering-based baselines. Notices that SPHDM can also be easily extended to some related tasks, such as phishing email detection, network intrusion detection, binary code cloning, *etc.*, by performing structural similarity detection.

In particular, the key contributions are listed as follows:

- A method for analysing the homology of phishing webpages is proposed. Based on the structural similarity of phishing webpages belonging to the same family, two kinds of structural features are extracted to form webpage representations, including DOM (Document Object Model) structure and Class attribute corresponding to CSS (Cascading Style Sheets) styles, and similarity calculation method is designed. The method provides new ideas for homology analysis of phishing webpages.
- In order to speed up the detection of webpages to be tested, a fingerprint generation algorithm is proposed. Through twice compression, fingerprints are generated for each cluster and webpage to be tested, simplifying the comparison and classification of webpages.
- Further, four experiments on the SPHDM are conducted from different aspects. The results show that the classification performance is good.

The paper is organized as follows. In "Related Works", we present related works on phishing webpage detection. Then, the framework and the detailed process of SPHDM is described in "Proposed Method". In "Experimental Results and Analysis", the performance of the SPHDM is evaluated. Finally, we conclude the paper and discuss future works.

## RELATED WORKS

The typical methods for detecting phishing webpages as classification problems include blacklisting, heuristics, machine learning, and deep learning (*Liang et al., 2016*; *Xiang et al., 2011*; *Moghimi & Varjani, 2016*; *Raj & Vithalpura, 2018*; *Rao & Pais, 2020*; *Aleroud & Zhou, 2017*; *Rao & Pais, 2019*; *Bahnsen et al., 2017*; *Feng, Zou & Nan, 2019*; *Yang, Zhao & Zeng, 2019*; *Feng et al., 2020*). There have been a lot of researches and they have reached a relatively mature stage. At the same time, there are relatively few studies that regard phishing webpage detection as a clustering problem, which can be divided into visual similarity-based method and structural similarity-based method.

The method based on visual similarity starts from the visual characteristics of webpages and can effectively identify brand attacks. The earliest representative research result is the literature *Liu et al. (2005)*. This research compares the similarity of phishing webpages with the imitated original webpages from visual features such as text, style and layout of webpages. Inspired by literature *Liu et al. (2005) etc.*, CANTINA+ (*Xiang et al., 2011*) enhanced the detection effect of phishing webpages by analysing specific tags in the DOM tree by adopting a classification method. With the development of researches, *Li et al. (2019)* introduces a visual similarity matching algorithm based on the rendering tree constructed through the DOM and CSS rule trees. Generally speaking, comparing the similarity of webpages from a visual perspective requires a large amount of image calculations, and the complexity and resource consumption are high.

The work of phishing detection based on structural similarity is not limited to brand attacks. It usually aims at discovering the family of phishing webpages by clustering webpages based on the DOM structure of webpages. For webpages, DOM of HTML (HyperText Markup Language) is a kind of semi-structured document consisted of HTML tags and their attributes, and as the skeleton structure of webpages can provide effective clues for structural similarity. Typical research includes the literature *Rosiello et al. (2007)*, which compares the similarity of the DOM tree extracted from the HTML source code through simple tag comparison and isomorphic subgraph recognition. Among them, the tag comparison uses a tag-by-tag comparison method, which results in low efficiency, while the isomorphic subgraph method requires a large amount of calculation. In order to improve the comparison efficiency, some works map the DOM structural features into simplified vectors. For example, the HTMLTagAntiPhish method proposed by *Zou et al. (2016)*. Only encodes the representative tag sequences in the DOM, and measures the similarity according to the alignment scores between different sequences. *Cui et al. (2017)* proposed a method TC (Tag Counting) to measure the similarity between webpages by counting the frequency of tags and generate a fixed-length tag vector for each webpage. Considering that shallow nodes are more important than deep nodes in the DOM structure, literature *Feng & Zhang (2018)* uses the Hierarchical Distance (HD) of hierarchical DOM tags to characterize the structural characteristics of the DOM, thereby measuring differences between webpages. The above-mentioned typical webpage structure similarity calculation methods based on string (*Rosiello et al., 2007*), symbolic (*Zou et al., 2016*; *Cui et al., 2017*), tree (*Feng & Zhang, 2018*) and figure (*Rosiello et al., 2007*) still have problems of low precision and low efficiency.

Table 1 summarizes and compares the clustering-based methods mentioned above.

The research on the structural similarity comparison of webpages is very meaningful but not sufficient until now. Both the depth and breadth of the research need to be improved. In essence, the key step of webpage similarity calculation is webpage representation. The way of representing webpages determines the degree and granularity of information extraction, which in turn affects the accuracy and efficiency of similarity comparison. Based on the existing structure-oriented webpage similarity comparison method, considering the characteristics of the generation and evolution of phishing webpages and integrating structural related features, this paper designs a new webpage

**Table 1 Clustering-based phishing detection methods.**

| Research | Type | Research object | Innovation | Drawbacks |
|---|---|---|---|---|
| *Liu et al. (2005)* | Visual similarity | Block and layout | First work of visual similarity | For brand attacks |
| *Li et al. (2019)* | | RenderLayer tree | RenderLayer tree built | High complexity |
| *Rosiello et al. (2007)* | | DOM | Tag comparison/Isomorphic subgraph recognition | Low efficiency/High computational load |
| *Zou et al. (2016)* | Structural Similarity | DOM | Sequence alignment | Only consider the representative tags |
| TC (*Cui et al., 2017*) | | DOM | Tag vector built | Based only on the frequency of tags |
| HD (*Feng & Zhang, 2018*) | | DOM | Distinguish importance of tags hierarchically | No CSS properties |

similarity measurement method to improve accuracy and efficiency of phishing webpages detection through webpage traceability. To address the drawbacks of existing researches, on the one hand, by proposing a new hierarchical tag vector construction method and considering Class style attributes besides the DOM tree, SPHDM optimizes the expression of structure; on the other hand, SPHDM designs an fingerprint algorithm to improve the detection efficiency by low computational load.

# PROPOSED METHOD

In this section, the problem statement of phishing detection is given firstly, and then the overall framework of SPHDM and its key technologies are gone into detail.

## Problem statement

The basic idea of our work is to regard the detection of phishing webpages as the clustering of webpages with the same or similar structure. Firstly, based on the hierarchical structure of webpages and other structural elements that affect page layout to establish a feature library for each phishing family, and to select a representative collection of phishing webpages to realize the traceability. Secondly, when an unknown webpage appears, its structural characteristics are extracted and compared with the most representative collection of phishing webpages to determine whether it belongs to a certain phishing family. The process of pairwise enumeration and comparison should be simplified to reduce the complexity of comparison and also weaken the influence of kit evolution on structural similarity.

## Proposed framework

Architecture of SPHDM is shown in Fig. 2. SPHDM is divided into two stages: modelling and prediction.

The input of the modelling is training set, includes benign webpages and phishing webpages. First, to extract the structural features of these webpages to construct representation vectors. Then, the webpages are clustered according to the similarity calculation method between vectors. Finally, according to the third-party blacklist library, each cluster is labelled as benign or phishing.
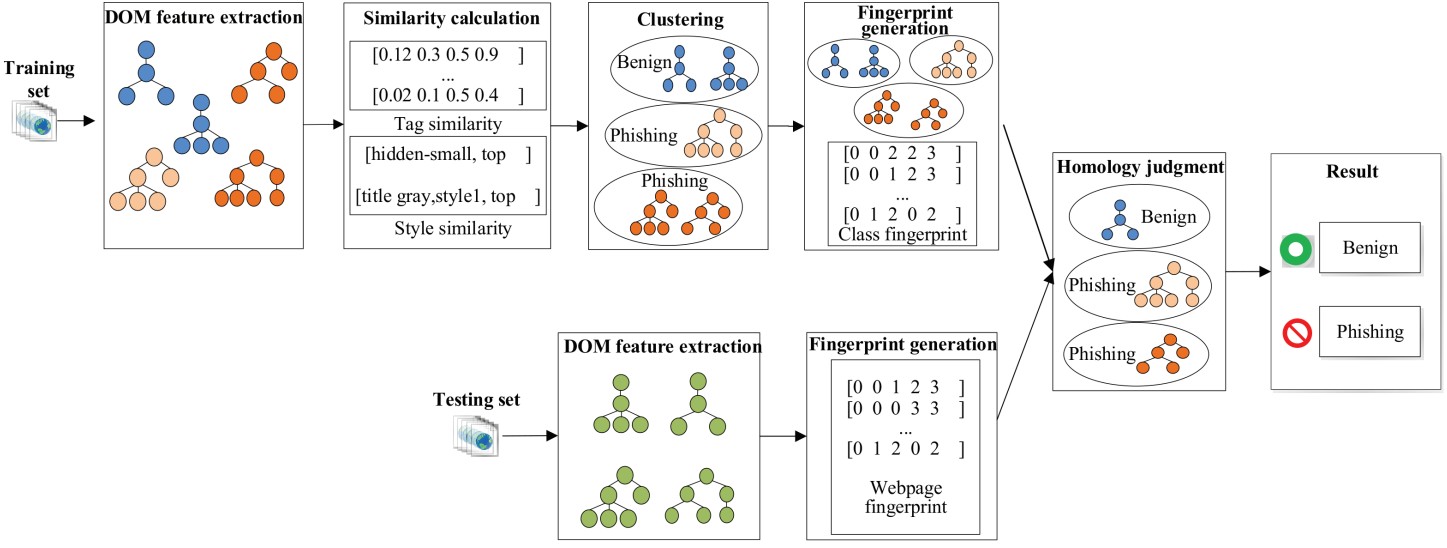

**Figure 2  Architecture of SPHDM.**                                     

In the prediction stage, the webpage to be tested and the webpages as centre of the clusters are represented as vectors by a fingerprint generation algorithm. Then the webpage to be tested is classified into a certain cluster according to the similarity between the vectors and has the same label to the cluster it belongs to.

## Phase 1 modelling

The key to tracing the source of phishing webpages through clustering is to measure the homology of webpages. The modelling stage includes three processes: structure feature extraction, webpage representation and clustering.

### Structural feature extraction

HTML documents are typical semi-structured documents, in which there is a nested relationship between tags, which reflects the hierarchical structure of the webpage and can be described by the DOM tree. At the same time, when using the kits to generate a number of webpages, attackers usually reuse CSS style, which results in the generated webpages using a consistent set of CSS properties. However, existing research has neglected the importance of CSS styles for webpage layout, so as a supplement to the DOM structure, this paper also extracts the Class attribute of CSS style as a structural feature.

### DOM structural characteristics

The DOM represents an HTML document as a tree structure with tags, attributes, and text nodes. In order to simplify the calculation, only the DOM tag tree is used to represent the HTML document, and attributes, text and comment nodes are ignored.

In order to highlight the hierarchical information of the tag tree, a structure table is constructed, which stores the hierarchy and tag sequence of the DOM tree in order, and the traversal strategy is depth first. Since the information is mainly within the element <body>, the part under <head> is not extracted. Figure 3 converts an HTML document

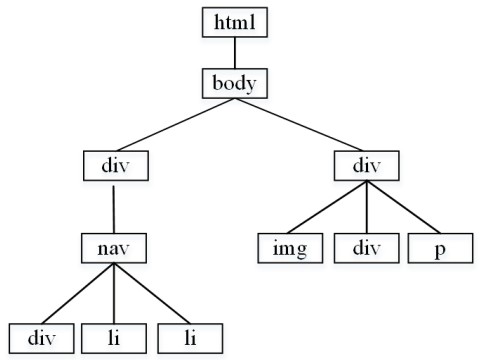

| Node ordering | Layer Number | Tag Name |
|---|---|---|
| 1 | 1 | html1 |
| 2 | 2 | body2 |
| 3 | 3 | div3 |
| 4 | 4 | nav4 |
| 5 | 5 | div5 |
| 6 | 5 | li5 |
| 7 | 5 | li5 |
| 8 | 3 | div3 |
| 9 | 4 | img4 |
| 10 | 4 | div4 |
| 11 | 4 | p4 |

**Figure 3 DOM tag tree and structure table.**

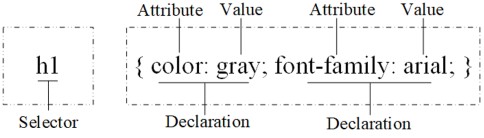

**Figure 4 CSS rule.**

```
<style type="text/css">
    h1.intro {color:blue;}
    p.important {color:red;}
</style>

    <h1 class="intro">My Header</h1>
    <p>This is a paragraph text.</p>
    <p class="important"> This is an important paragraph text.</p>
```

**Figure 5 CSS settings in DOM.**

into a corresponding DOM tag tree and stores it in the structure table. The resulting hierarchy tag sequence is [html1, body2, div3, nav4, div5, li5, li5, div3, img4, div4, p4], shown as Fig. 3.

*Class attribute characteristics*

CSS rules are used to formulate layout of webpages, which contain selectors and declaration information, as shown in Fig. 4. The declaration contains attributes and corresponding values.

Figures 5 and 6 show the CSS styles in the DOM of a certain webpage and the page after the corresponding styles are set. Because values of the attributes often change a lot, only the attributes are extracted as structural features. As shown in Fig. 5, the Class attribute set extracted is [intro, important].

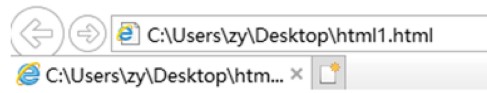

**Figure 6  HTML page using CSS settings.**         

## Webpage representation

Although the DOM hierarchy tag sequence and the Class attribute set are extracted, the commonly used sequence comparison method is not used for its complexity. Instead, we calculate its representation for each webpage by the hierarchical tag vector and the Class attribute vector, and through the vector similarity calculation to reduce the complexity of comparison between webpages.

### Hierarchical tag vector

In order to reduce the complexity of DOM tree comparison, it is necessary to simplify the representation of the tree-level tag sequence, and at the same time express the hierarchical characteristics of the webpage and distinguish the differences in the structural elements of the webpage, so as to improve the effect of feature expression. TF-IDF (Term Frequency-Inverse Document Frequency) is a statistical document representation method widely used in the field of information retrieval (*Khan et al., 2010*), in which the word vector is determined according to TF, and IDF is used to adjust the weight. But IDF cannot effectively express the importance of tags and their distribution. Therefore, we assign weights through the role of tags, and use improved TF-IDF to vectorize the DOM hierarchical tag sequence.

Suppose there are m types of hierarchical tags in a webpage set, this is $TagType = [tag_1, tag_2,\ldots,tag_m]$, using TF and IDF to determine the frequency and importance of tags with hierarchical information as follows:

$$TF_{ij} = \frac{|tag_j|^{p_i}}{\sum_{a=1}^{m} |tag_a|^{p_i}} \tag{1}$$

$$IDF_{ij} = \log\left(\frac{n}{1 + |p_i, tag_j \in p_i|}\right) \tag{2}$$

where in $TF_{ij}$ represents the ratio of the number of occurrences $|tag_j|^{p_i}$ of $tag_j$ in webpage $p_i$ to the total number of occurrences $\sum_{a=1}^{m} |tag_a|^{p_i}$ of all tags, namely, frequency; $IDF_{ij}$ is the importance of $tag_j$. Among them, $n$ represents the total number of webpages, and $|p_i, tag_j \in p_i|$ represents the number of webpages containing $tag_j$.

There are 117 commonly used HTML tags, can be roughly divided into three categories: layout-related tags, text-related tags, and other tags. Different tags have different effects on webpages. For example, layout-related tags have larger effects on the page layout, and text-related tags only affect the text display. The existing similarity comparison methods

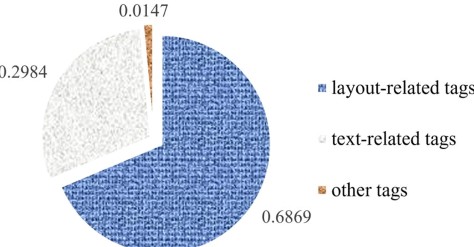

**Figure 7 Frequency of tag usage.**

for DOMs ignore this observation, but in our opinion, the use frequencies of tags can reflect the importance of the tag categories. Through the statistics of DOM trees from benign webpages collected, it is found that the use frequency of layout-related tags such as div, span, li, option, *etc.* is 7/3 times than that of text-related tags, showed in Fig. 7, so we set the weights for each category: layout-related tags are 7/3, text-related tags are 1, and other cases are 0.

After weighting, the hierarchical tag vector of $p_i$ is:

$$V_T^{p_i} = [z_{i1}, z_{i2}, \ldots, z_{im}] \tag{3}$$

where $z_{ij} = TF_{ij} * IDF_{ij} * weight$.

*Class attribute vector*

Class attribute is set type, and each set contains multiple attribute strings. Therefore, the sets of webpages with similar styles and layouts will have some common elements, so they can be directly embedded.

$$V_C^{p_i} = [ss_{i1}, ss_{i2}, \ldots, ss_{iq}] \tag{4}$$

In the Eq. (4), the attribute vector of webpage $p_i$ contains $q$ attributes.

### Similarity measurement

The tag vector and the Class attribute vector are the basis for similarity calculation. Firstly, the tag hierarchical representation matrix is used as input, and the tag difference is calculated by comparing the maximum dissimilarity. Secondly, the Dice coefficient is used to measure the Class attribute difference. Finally, the combined value of the two is used to measure the total difference of webpages to express the similarity between webpages.

Definition 1 Tag difference $(D_T)$

$$D_T(p_1, p_2) = \frac{DiffValue}{MaxValue} \tag{5}$$

$$DiffValue = \sum_{i=1}^{m} |z_{1i} - z_{2i}| \tag{6}$$

The tag vector of webpage $p_1$ is $V_T^{p_1} = [z_{11}, z_{12}, \ldots, z_{1m}]$ and the tag vector of webpage $p_2$ is $V_T^{p_2} = [z_{21}, z_{22}, \ldots, z_{2m}]$, *MaxValue* is the maximum value of *DiffValue*.

Definition 2 Class attribute difference $(D_C)$

$$D_C(p_1, p_2) = 1 - \frac{2 * |V_C^{p_1} \cap V_C^{p_2}|}{|V_C^{p_1}| + |V_C^{p_2}|} \tag{7}$$

where "| |" means getting the number of elements in a set.

Definition 3 Total difference between webpages (D)

$$D(p_1, p_2) = \alpha * D_T(p_1, p_2) + \beta * D_C(p_1, p_2) \tag{8}$$

In order to distinguish the influence of structure and attribute on similarity calculation, set the tag vector importance factor $\alpha$ and the Class attribute vector importance factor $\beta$ separately, where $\alpha + \beta = 1$. It can be seen from Eq. (8) that if the total difference value is larger, the two HTML pages are less similar.

Figure 8 provide an illustrative example for above process.

### Clustering

The partition-based method is a typical clustering method. It often divides the dataset into $k$ groups, each of which represents a category, such as k-means and k-medoide algorithm (*Modak, Chattopadhyay & Chattopadhyay, 2020*). However, such methods need to set the number of clusters in advance. Since the number of phishing clusters cannot be determined in advance, they cannot be directly adopted. So, a k-cluster algorithm is proposed to determine the number of clusters according to actual situation. Firstly, the selection of the initial centre set is performed. After $k$ initial centre points are obtained, the webpage set is divided iteratively using the k-medoide clustering method until the clustering results no longer change. The steps for selecting the initial cluster centre are as follows:

1. Set a webpage selected randomly from the webpages as the initial centre point.
2. Use Eq. (8) to calculate the difference between other webpages and the central point, group webpages and central points smaller than the threshold $\theta$ into one cluster, and find a new central point which is the closest webpage to the mean of this cluster.
3. Randomly select a webpage from the webpages outside existed cluster(s) as the initial centre point and repeat step 2.
4. Repeat step 3 until the clustering is completed, get $k$ initial centre points.

Although random webpage is selected at the beginning, a reasonable one will be got iteratively. This will eliminate the influence of random initial cluster centre on the number of clusters. After the clustering is completed, phishing clusters with similar structures are obtained, and the clusters are labelled by known webpage labels.

### Phase 2 prediction

The webpage to be tested is usually classified according to its distance from the centre of clusters, and then the webpage is marked according to label of the cluster. The problem is that when there are many clusters, the computational efficiency is low. In order to improve computational efficiency, a Fingerprint Generation (FG) algorithm is proposed to generate fingerprints for webpages by extracting their key structural features, so as to realize the fuzzy and fast detection.

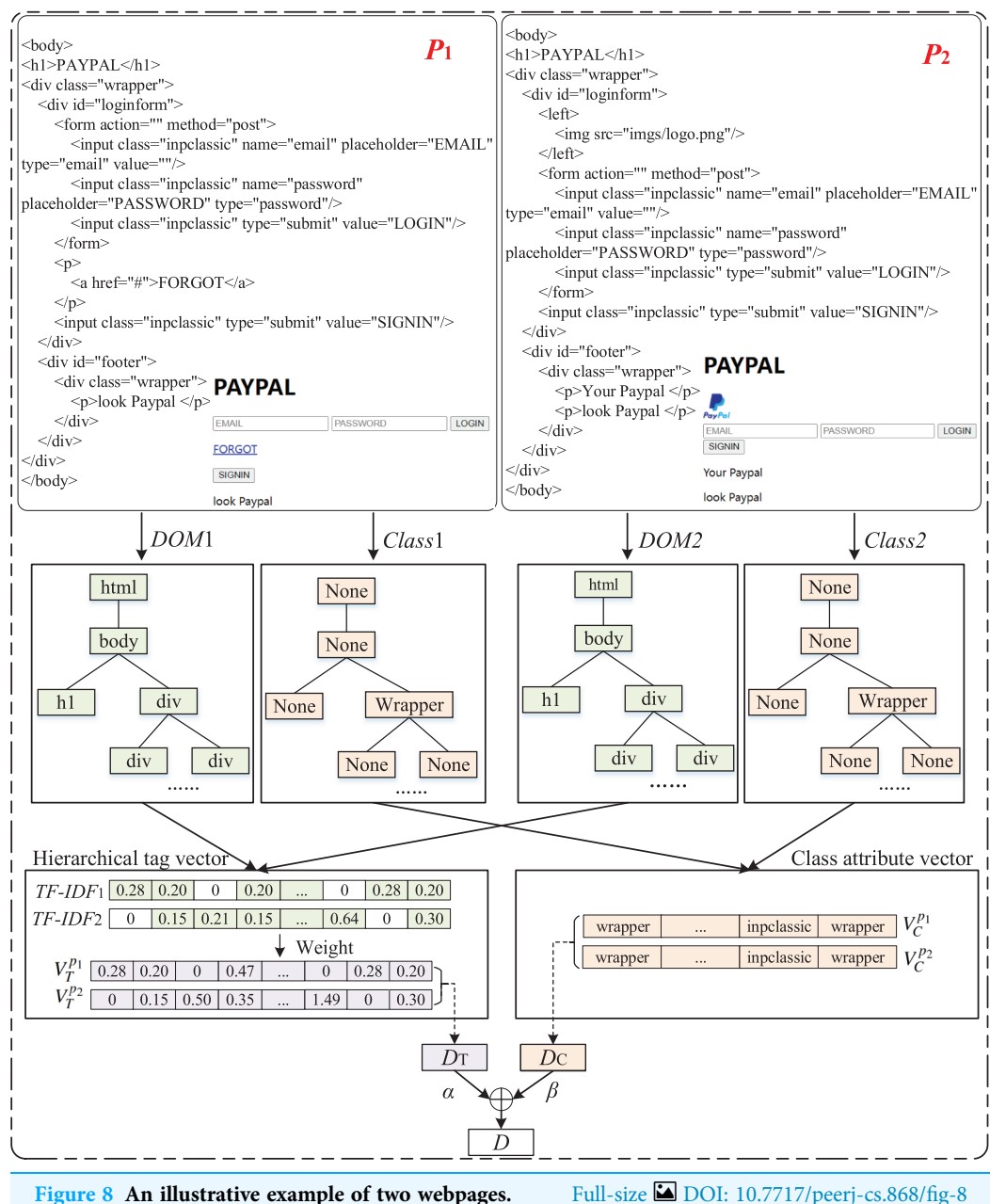

**Figure 8** **An illustrative example of two webpages.**

### Fingerprint generation

A fingerprint can be understood as a short fixed-length character string. In order to generate fingerprints, the original webpage needs to be compressed. FG is divided into two stages, *FGµ* and *FGη*.

### FGµ stage

The tag sequence is read sequentially, and the initial fingerprint is generated using the LZ78 compression algorithm (*Barua et al., 2017*). That is, if a certain HTML tag has not appeared, the code is 0; if it has appeared, the longest prefix record of the tag is searched,

| Hierarchy | Tags |
|:---:|:---:|
| 1 | html |
| 2 | body |
| 3 | h1 div |
| 4 | div div center |
| 5 | center form p input div img img |
| 6 | img img img input input input  a p p |

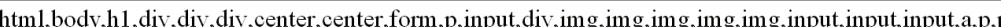

html,body,h1,div,div,div,center,center,form,p,input,div,img,img,img,img,img,input,input,input,a,p,p

**Figure 9** **Tag sequence.**

and the code is value of the longest prefix index. A code table is constructed to record the above information.

By analysing DOM trees, it is found that the shallow nodes of the trees have greater impact on the webpage structure. If two webpages are not similar, the shallow tags will be quite different; but if two webpages are similar, the shallow tags are similar, but the deeper the level, the difference bigger. That is to say, the in-depth information of the DOMs will interfere with the expression of the structures, so exact comparison between DOMs is not desirable. Set threshold $l$ for the length of fingerprint to limit the length of the output fingerprint and weaken the influence of the deep nodes of the DOM structure.

*FGη stage*
Perform second compression based on the initial fingerprint, that is, convert the repetitive codes in the initial fingerprint into codewords and the number of occurrences to form the final webpage fingerprint sequence *newFP = FGη(FinP)*. Specifically, the rules are as follows:

- If *there is no continuous repetition of the code*: only the code is added in *newFP*;
- If *the number of consecutive occurrences of the code* ≥ 2: add the code and the corresponding number of repetitions to *newFP*.

Here is an example of the above fingerprint generation process. Assume that the tag sequence of a webpage is shown in Fig. 9.

According to the tag sequence, give serial number to the tags, shown in Fig. 10. Read the label in sequence, if a tag is appeared for the first time, give it a serial number; otherwise, if an existing tag occurs, continue to read the next tag for combination judgment. If the combination did not appear before, connect the tag below the existing one and number it.

After all tags are numbered, the code table will be constructed, shown in Fig. 11.

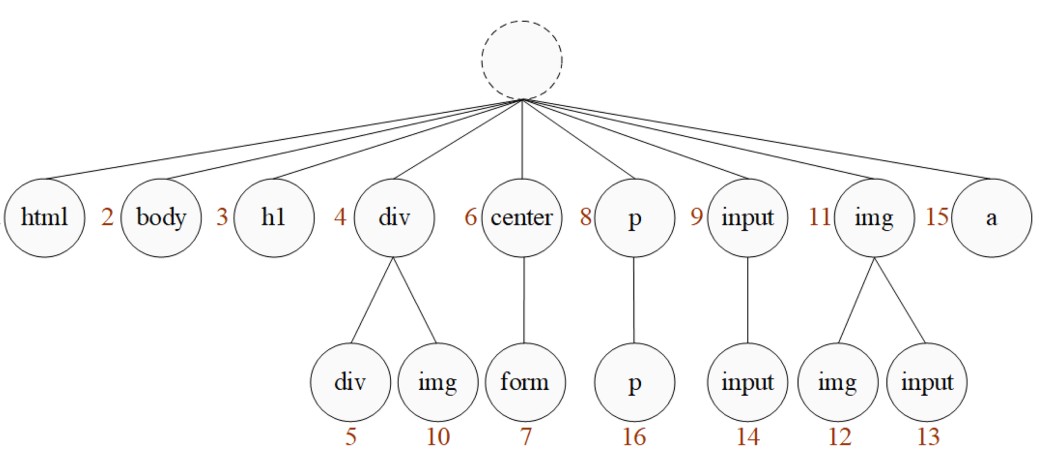

**Figure 10 Numbered tags.**

| Number | The longest prefix | Output | Index of the longest prefix |
|--------|--------------------|--------|-----------------------------|
| 1  | -      | html          | **0** |
| 2  | -      | body          | **0** |
| 3  | -      | h1            | **0** |
| 4  | -      | div           | **0** |
| 5  | div    | **div** div   | 4 |
| 6  | -      | center        | 0 |
| 7  | center | **center** form | 6 |
| 8  | -      | p             | **0** |
| 9  | -      | input         | **0** |
| 10 | div    | **div** img   | 4 |
| 11 | -      | img           | 0 |
| 12 | img    | **img** img   | **11** |
| 13 | img    | **img** input | **11** |
| 14 | input  | **input** input | 9 |
| 15 | -      | a             | 0 |
| 16 | p      | **p** p       | 8 |

**Figure 11 Code table.**

According to FG algorithm, after the *FGμ* stage, use the code table to output the number of the longest prefix index, *FGμ(Seq)* = *FinP* = [0, 0, 0, 0, 4, 0, 6, 0, 0, 4, 0, 11, 11, 9, 0, 8]. Then, after the *FGη* stage, the longest prefix index column is further converted, so *FGη(FinP)* = *newFP* = [0, **4**, 4, 0, 6, 0, **2**, 4, 0, 11, **2**, 9, 0, 8], this is the final fingerprint generated after twice compressions.

To improve the comparison efficiency, unlike the modelling stage, the Class attribute is not considered in the fingerprint generation process.

In some special circumstances, the fingerprints may be the same even facing different webpages. Because in *FGμ* stage, only partial front DOMs are taken to generate *l* bytes fingerprint. If two webpages have same part of their DOMs, the fingerprints will be the same. *FGη* stage only convert repetitive codes into codewords and there is no information loss, so the same fingerprints created in *FGμ* stage will keep same after that. But for two different clusters, the fingerprints will be different because centre points of two clusters have different low lever DOMs, or they will be in one cluster. This is what we need: to simplify the comparison of webpages, remove the redundancy of internal information while realize the fuzzy and fast detection.

### Webpage classification

After generating fingerprints for the webpage to be tested, make a bitwise comparison to the fingerprints of the various clusters, and classify the webpage into the cluster with the difference value $D$ less than the threshold $\varphi$, and make it labelled according to the category of the cluster.

## EXPERIMENTAL RESULTS AND ANALYSIS

In order to verify the validity of the SPHDM model, two sets of experiments were designed to try to answer the following questions:

- Question 1: Can SPHDM accurately detect phishing webpages?
- Question 2: Does SPHDM improve detection efficiency?

### Experiment preparation

#### Experimental environment and dataset

The experimental development environment is shown in Table 2.

The webpages used in the experiments come from Internet. Among them, the benign webpage collection is from Alexa. Alexa is a website maintained by Amazon that publishes the world rankings of websites. We collect webpages in the top list provided by Alexa which are considered as benign webpages. After filtering out invalid, error, and duplicate pages, 10,922 benign webpages are collected.

The phishing webpage collection comes from PhishTank.com. PhishTank is an internationally well-known website which collects suspected phish submitted by anyone, verifies it according to whether it has a fraudulent attempt or not, and then publish a timely and authoritative list of phishing webpages for research. Due to the short survival time of phishing webpages, we collected totally 10,944 phishing webpages listed on

**Table 2 Development environment.**

| Operating system | Processor | Memory | Development environment | Development language |
|---|---|---|---|---|
| Windows 10 | Intel Core i5-3337U | 4 GB | Eclipse | Python2.7 |

**Table 3 Evaluation indicators.**

| Evaluation indicator | Calculation formula |
|---|---|
| Precision | TP/(TP + FP) |
| TPR | TP/(TP + FN) |
| FPR | FP/(TN + FP) |

PhishTank every day from September 2019 to November 2019, and processed the webpages that did not meet the grammar rules.

### Evaluation indicators

To summarize various evaluation indicators in the literatures, the most commonly used are the following: Precision, True Positive Rate (TPR) and False Positive Rate (FPR), and their calculation formulas are shown in Table 3.

Among them, TP (True Positive) denotes the number of phishing webpages correctly classified as phishing webpages, FP (False Positive) denotes the number of benign webpages classified as phishing webpages, TN (True Negative) denotes the number of benign webpages classified as benign webpages, and FN (False Negative) denotes the number of phishing webpages classified as benign webpages.

### Baselines

In SPHDM, although modelling and prediction are highly related tasks, they are often processed and solved independently in practical applications, so their effects are also verified separately in experiments.

For the modelling part, since the starting point is structural similarity of DOM, the typical methods are compared, which mainly include tree edit distance (ED (*Alpuente & Romero, 2010*)) method, tag frequency statistics (TC) method, and hierarchical distance (HD) method. In addition, it is compared with the traditional TF-IDF to illustrate the advantages of the improved TF-IDF similarity.

In the prediction part, it is compared with the typical web fingerprint generation algorithm Simhash (*Charikar, 2002*) and the encoding compression algorithm Huffman (*Henzinger, 2006*).

Notice that classification-based methods are not compared because SPHDM is a clustering-based method.

## Experimental evaluation

### Experiment 1

In order to evaluate the effectiveness of SPHDM, Experiment 1 compares SPHDM with the classic phishing webpage detection method based on structural similarity.

**Table 4 Selection of parameters $\alpha$ and $\beta$.**

| Tag coefficient $\alpha$ | Class coefficient $\beta$ | TPR/% | FPR/% | Precision/% |
|---|---|---|---|---|
| 1.0 | 0.0 | 90.75 | 0.22 | 99.8 |
| 0.9 | 0.1 | 90.10 | 0.05 | 99.9 |
| 0.8 | 0.2 | 91.12 | 0.10 | 99.9 |
| 0.7 | 0.3 | 90.54 | 0.25 | 99.7 |
| 0.6 | 0.4 | 89.36 | 0.55 | 99.4 |
| 0.5 | 0.5 | 90.90 | 0.98 | 98.9 |
| 0.4 | 0.6 | 90.35 | 3.24 | 96.5 |
| 0.3 | 0.7 | 91.26 | 8.12 | 91.8 |
| 0.2 | 0.8 | 91.74 | 8.62 | 91.4 |
| 0.1 | 0.9 | 91.34 | 8.60 | 91.4 |
| 0 | 1.0 | 91.78 | 8.62 | 91.4 |

**Table 5 Results under different $\theta$.**

| Threshold $\theta$ | TPR/% | FPR/% |
|---|---|---|
| 0.05 | 88.63 | 0.02 |
| 0.1 | 91.12 | 0.10 |
| 0.2 | 92.80 | 3.73 |
| 0.3 | 93.86 | 10.63 |
| 0.4 | 98.98 | 91.69 |
| 0.5 | 100.00 | 100.00 |
| 0.6 | 100.00 | 100.00 |

*Parameter setting for SPHDM*

In order to select suitable parameters, the experiment adjusts the two parameters $\alpha$ and $\beta$ in Eq. (8), shown in Table 4, and selects the best set of parameters.

From Table 4, as $\alpha$ decreases and $\beta$ increases, TPR fluctuates between 89.36%–91.78%. When $\alpha \leq 0.9$ and $\beta \geq 0.1$, FPR gradually increases, while Precision gradually decreases. Since the results are better when TPR and Precision take high values and FPR takes low values, it is obvious that $\alpha = 0.9$ and $\beta = 0.1$ are appropriate. Therefore, the experiment finally chooses the tag coefficient $\alpha = 0.9$ and the Class coefficient $\beta = 0.1$. This is in line with the actual situation. In webpages, the DOM reflects the global information of the webpage structure, while the Class attribute only reflects the detailed information, and the amount of the Class attribute is relatively small compared to the DOM, so the effect of the DOM tree is far greater on structure than the Class attribute.

The key to clustering is to find the optimal threshold $\theta$. The larger the value of $\theta$, the looser the clustering restriction, so the FPR will increase; on the contrary, the stricter the clustering restriction, the lower the FPR will be. The results under different $\theta$ are shown in Table 5.

It can be concluded from Table 5 that as the threshold gradually increases, both TPR and FPR are increasing. Especially when the threshold is greater than 0.3, the value of

**Table 6 Clustering results.**

| Method | Number of clusters | Clustering time/s | TPR/% | FPR/% |
|--------|-------------------|-------------------|-------|-------|
| ED | 2,048 | 6,818 | 84.25 | 1.91 |
| TC | 1,701 | 3,631 | 86.90 | 0.29 |
| TF-IDF | 1,700 | 4,503 | 88.89 | 0.73 |
| HD | 1,655 | 1,917 | 89.23 | 1.22 |
| SPHDM | 1,598 | 4,925 | 90.10 | 0.05 |

FPR rises rapidly. When $\theta$ is 0.1, TPR and FPR reach a good compromise. Therefore, set $\theta$ to 0.1.

*Detection effect*

Table 6 shows the number of clusters and effects after the execution of each method under the above parameters.

It can be seen from Table 6 that SPHDM has the best effect. ED is based on edit distance. In the case of complex webpage layout, it causes more mismatches, so the effect is worse than other methods. On the other hand, TC is based on tag frequency. When the webpage layout is complex, the similarity calculation result is better; but for the webpage with simple structure, because the total number of tags is small, the discrimination ability of tag frequency is low. TF-IDF weights the word frequency, which weakens the less influential tags to a certain extent, so the TPR is improved. HD not only compares the tags, but also considers the level of tags, so the overall performance is better. However, for webpages with strong homology, in order to improve the comparison efficiency, HD only extracts shallow tags for calculation, which has caused certain misjudgements. SPHDM combines tags and style attribute in the page structure, comprehensively considers the tag frequency, tag category and weight, and can better reflect the structural characteristics of webpages. Therefore, TPR and FPR have reached the optimal results.

From the perspective of the number of clusters, ED has the largest number of clusters due to its strict matching mode. SPHDM has the least number of clusters because it focuses on expressing homology and can better classify webpages with similar overall structures but small differences into one category. When more webpages are collected and used, the detection effect will be more accurate, because more cluster will be found.

In addition, the clustering of ED takes the longest time because it requires bitwise comparisons between sequences, while others are all based on statistical methods. TC and HD are simpler statistical method than TF-IDF. Comparing to TF-IDF, SPHDM considers both structure and semantics, and takes the CSS information into account, so needs longer clustering time.

*Cluster analysis*

By observing the clustering results, we find that phishing webpages mostly target at the mailboxes of some famous websites. The title information of the phishing webpages is extracted and listed in Table 7 together with the imitation targets.

| Table 7 Brands imitated frequently by phishing webpages. | |
|---|---|
| **Title of phishing webpages** | **Target brands** |
| only the best-yahoo | Yahoo |
| secure login | Mail box |
| google docs | Google |
| dropbox | Dropbox |
| yahoo-login | Yahoo |
| Login-PαyPαl | Paypal |

| Table 8 The effect of fingerprints with different length. | | |
|---|---|---|
| **Length $l$** | **TPR/%** | **FPR/%** |
| 10 | 98.10 | 5.23 |
| 15 | 98.23 | 4.72 |
| 20 | 90.28 | 0.27 |
| 25 | 88.08 | 0.09 |
| 30 | 88.71 | 0.08 |
| 35 | 89.86 | 0.06 |
| 40 | 87.76 | 0.06 |
| 50 | 87.68 | 0.05 |
| 60 | 87.65 | 0.06 |
| 70 | 87.61 | 0.05 |
| 80 | 86.59 | 0.05 |

It can be seen that the title of the phishing webpage basically corresponds to the target brand, which is easy to be confused in visual effect. It is worth noting that the same target brand often has many corresponding phishing webpages, but these phishing webpages are different in structure and created by different toolkits, so they belong to different clusters.

### Experiment 2

Experiment 2 conducted an experimental analysis on the efficiency improvement of the prediction stage in SPHDM.

*Parameter setting*

Table 8 shows the detection effect of webpage fingerprints with different lengths.

It can be observed from Table 8 that when the length of the fingerprint is increased to a certain extent, the effect becomes not very significant, but on the contrary, it will increase the comparison time. When $l = 35$, the result of is better than other values, so set $l = 35$. Notice that FG is basically a data compression algorithm, when the original data is long enough, the compressed result will be valid. In general, the DOM tree of a webpage is composed of a large number of tag sequences, which is enough to generate a 35-byte fingerprint.

**Table 9 Results under different φ.**

| Threshold φ | TPR/% | FPR/% | Precision/% |
|---|---|---|---|
| 0.1 | 87.94 | 0.03 | 99.9 |
| 0.2 | 89.86 | 0.06 | 99.9 |
| 0.3 | 89.84 | 0.19 | 99.7 |
| 0.4 | 89.58 | 0.70 | 99.2 |
| 0.5 | 91.26 | 1.93 | 97.9 |
| 0.6 | 92.73 | 9.39 | 90.8 |
| 0.7 | 98.76 | 82.55 | 54.5 |
| 0.8 | 100.0 | 99.94 | 50.0 |
| 0.9 | 100.0 | 100.0 | 50.0 |

**Table 10 Detection results.**

| Method | TPR/% | FPR/% |
|---|---|---|
| SPHDM | 90.10 | 0.05 |
| SPHDM+Huffman | 86.37 | 0.20 |
| SPHDM+Simhash | 89.88 | 14.81 |
| SPHDM+LZ78 | 89.56 | 0.10 |
| SPHDM+FG | 89.86 | 0.06 |

Table 9 shows the influence of the fingerprint difference threshold $\varphi$ on the detection result. This parameter is the classification condition of the unknown webpage. It can be seen from Table 9 that FPR increases with the increase of the threshold. Therefore, considering TPR and FPR, set $\varphi = 0.2$.

*Detection effect*

When detecting the webpage to be tested, Table 10 shows the effect of combining SPHDM with different compression methods.

Observing Table 10, it can be seen that in the prediction phase, directly using the features of the modelling phase, the detection effect is the best. SPHDM+FG adopts FG to collect fingerprints and compare fingerprints during prediction. Although the effect is slightly reduced, but the gap is very small. The problem of using LZ78 after clustering is that the information in front of the coding table has no index for reference at first, and most of it will be 0, which causes the number of identical elements to be increased when comparing fingerprints, which affects the similarity. Twice compression of FG weakens the influence of such situations, so the result is slightly better. TPR of Simhash is quite ideal because it uses the MD5 algorithm to hash webpages, which can achieve a high degree of similarity for similar webpages. But at the same time, Simhash is very sensitive to differences in webpages, and can distinguish slightly different hash values, so fuzzy comparison cannot be achieved well. Huffman only considers the feature frequency, so the discrimination effect is poor.

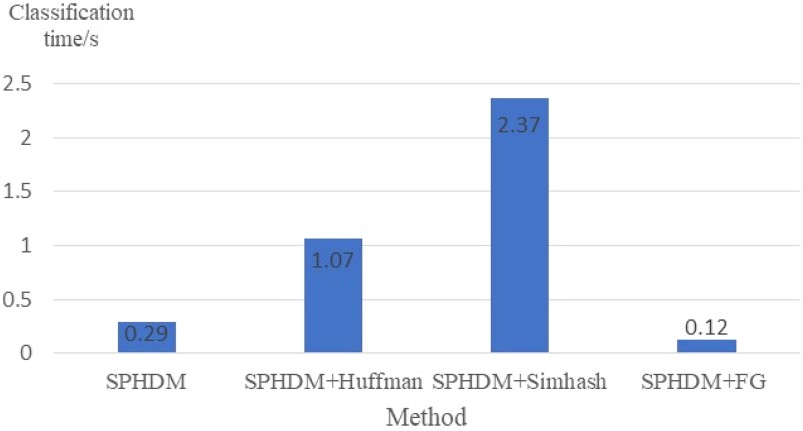

**Figure 12 Comparison of average webpage classification time.**

*Detection efficiency*

FG is proposed to improve the prediction efficiency. Figure 12 shows the average webpage classification time under different methods, which refers to the average time, including fingerprint generation time.

It can be seen from Fig. 12 that Simhash takes the longest time and FG takes the shortest time, that is, the efficiency is the highest. This is because fingerprint generation process of Simhash is more complicated. It needs to perform pre-processing operations such as word segmentation on the tag sequence, and then perform operations such as hash calculation, feature weighting, accumulation and merging, and fingerprint dimensionality reduction. On the other hand, FG can accelerate the comparison process due to the simple calculation but fixed-length fingerprint.

Notice the SPHDM only needs to cluster once to capture the basic structure information and represent a phishing family. The newly generated phishing webpages still have high structural similarity with some existing phishing webpages over a period of time if they belong to the same family. But for the classification methods, because feature evolution may make feature learning invalid, adjustments for classification model are required, which increases the time required for training and the difficulty of deployment, so SPHDM could be more efficient from this perspective.

Through the above two sets of experiments, it can be seen that from the perspective of homology, SPHDM is feasible to extract the structural features of webpages for family tracing, and FG can effectively improve the detection efficiency.

## CONCLUSIONS

From the perspective of whether webpages have homology, a cluster-based phishing webpage detection model SPHDM is proposed. The model combines the DOM hierarchy tags and the Class attribute corresponding to the CSS style to express the structural characteristics of webpages, realizes the traceability of the phishing kits through clustering, and proposes FG algorithm to accelerate the classification of unknown pages. Experiments

show that compared with the existing phishing webpage detection methods based on structure clustering, SPHDM has a good detection effect and high efficiency.

The research hypothesis of the paper is to treat the DOM structure of a webpage as a tree. More generally, if the DOM structure is regarded as a graph, then the similarity comparison between two webpages is a similarity comparison problem between the two graphs, and the underlying scientific problem is graph matching or network alignment. At present, the research of graph neural network (GNN) is in full swing. If GNN can be used to solve scientific problems such as graph matching and network alignment, and establish a more effective detection model, it will be important to improve the effect of phishing webpage detection and homology analysis.

### Funding
This work was supported by the Shaanxi Provincial Natural Science Foundation Project (Nos. 2020JM-533 and 2020JM-526) and the Chinese Postdoctoral Science Foundation (No. 2020M673446), and there was no additional external funding received for this study. The funders had no role in study design, data collection and analysis, decision to publish, or preparation of the manuscript.

### Grant Disclosures
The following grant information was disclosed by the authors:
Shaanxi Provincial Natural Science Foundation Project: 2020JM-533 and 2020JM-526.
Chinese Postdoctoral Science Foundation: 2020M673446.

### Competing Interests
The authors declare that they have no competing interests.

### Author Contributions
- Jian Feng conceived and designed the experiments, authored or reviewed drafts of the paper, and approved the final draft.
- Yuqiang Qiao performed the experiments, analyzed the data, performed the computation work, prepared figures and/or tables, and approved the final draft.
- Ou Ye performed the computation work, authored or reviewed drafts of the paper, and approved the final draft.
- Ying Zhang conceived and designed the experiments, performed the experiments, analyzed the data, performed the computation work, prepared figures and/or tables, and approved the final draft.

### Data Availability
The complete code and data are available in GitHub: https://github.com/qiaodaben/SPHDM/.

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
