# Peer review of "Detecting phishing webpages via homology analysis of webpage structure"

_PeerJ Computer Science, doi:10.7717/peerj-cs.868_

## Round 0.1 · original submission · Major Revisions

Please read the comments carefully.

Reviewer 1 ·

Basic reporting

- The problem statement should be included implicitly in the introduction section. No need to repeat it again in the Method section.
- It is mentioned that (in abstract and other section): "the proposed method can accurately locate the family of phishing webpages and can detect phishing webpages efficiently".
However, the proposed method is based on clustering which has high computational cost. Please justify how the proposed method is more efficient?
- is it possible to discuss the benefits of this kind of detection method comparing to using ML methods?

Experimental design

- The experimental design is described well. However, the time of detection should discuss the clustering time which is necessary to perform this method.
Also discuss the cost/time of generating the fingerprint?

- Will the fingerprint will be the same even you use different websites in the testing? When it should be updated? Will the detection rate will be enhanced if we use more websites in the training? Discuss this.

Validity of the findings

- The validation (not evaluation) of the proposed method is not clear. Example: How you ensure that the proposed fingerprint is valid?

The authors compared the proposed method using different parameters (for the same method). Is it possible to compare the performance of this proposed method with other previous methods to show the enhancements obtained?

Additional comments

- The best values in all tables should be highlighted (bold) to make the comparison easy.

Reviewer 2 ·

Basic reporting

Certain sections of the paper require major revision before the manuscript can be considered for publication.

Experimental design

Necessary references to the existing techniques should be provided in the experimental result section. Moreover, a comparative analysis of the proposed technique needs to be carried out with other techniques proposed in the literature to validate the effectiveness of the proposed model

Validity of the findings

No Comment

Additional comments

Some sections of the paper require proofreading to fix grammatical and typographical errors.

Annotated reviews are not available for download in order to protect the identity of reviewers who chose to remain anonymous.

---

## Round 0.2 · accepted · Accept

Thank you for your submission to PeerJ Computer Science. Your manuscript has been Accepted for publication. Congratulations!

Reviewer 2 ·

Basic reporting

No comment

Experimental design

No comment

Validity of the findings

No comment